

# Effects of *Undaria pinnatifida*-derived brown algae polysaccharide (UPS) on the nutritional composition, digestive capacity, immune performance and intestinal microbiota of juvenile sea cucumber (*Apostichopus japonicus*)

Jinghao Dong[1], Hailong Ma[1], Kuimei Liu[1], Wei Zhou[2], Suya Liu[1], Yongjun Sun[3], Wenming Ju[3] and Shanshan Wang[1]

[1] Rongcheng Campus, Harbin University of Science and Technology, Weihai, Shandong, China
[2] Rongcheng Hongde Marine Biotechnology Co., Ltd., Weihai, Shandong, China
[3] Homey Group Co., Ltd., Weihai, Shandong, China

Corresponding author
Shanshan Wang,
sswang@hrbust.edu.cn

## ABSTRACT

**Background:** Brown algae polysaccharides (BAPs), derived from marine brown algae, represent bioactive macromolecules with potential functional feed applications as novel feed additives for improving the health and nutritional quality of aquatic animals. Previous studies have shown that BAPs possess antioxidant, anti-inflammatory, immunomodulatory, antimicrobial, and antiviral activities. BAPs extracted from *Undaria pinnatifida* (UPS) contain kinds of BAPs such as alginate, mannitol, fucoidan and fucoheterosaccharides but there are few studies on the combined effects of these BAPs.

**Methods:** This study investigated the effects of UPS by supplementing juvenile sea cucumbers with 0%, 0.5%, 1%, 1.5%, 2%, and 3% UPS (polysaccharide/sea cucumber biomass, W/W). After 60 days feeding, the nutritional composition of body wall, digestive capacity and immune performance were analyzed. The diversity of intestinal microbiota in sea cucumber was analyzed using 16S rRNA gene sequence amplification technology to elucidate the effects of UPS supplementation on the composition and function of intestinal microorganisms.

**Results:** It was demonstrated that UPS supplementation significantly increased the nutritional content of the juvenile sea cucumber body wall although growth performance remained unchanged. The polysaccharide content peaked at 1% supplementation of UPS, which was 1.6-fold higher than that of the control group. Moreover, UPS also enhanced intestinal digestive enzyme activity such as cellulase or lipase activity, which was maximized at 1% or 3% supplementation with 5.8-fold and 1.6-fold higher than that of the control group. Additionally, UPS could improve immune performance of juvenile sea cucumber by significantly elevating superoxide dismutase activity (T-SOD). It was worth noting that low UPS concentrations (1% and 1.5%) reduced malondialdehyde (MDA) content while high concentrations (2% and 3%) increased it, indicating that low UPS supplementation may had a better effect on immune performance. 16S rRNA gene sequencing revealed that UPS supplementation reduced pathogenic *Ralstonia* populations. Function analysis
showd that UPS may modulate metabolic pathways related to nitrotoluene and carbon source utilization. In summary, *Undaria pinnatifida*-derived BAP especially at low concentrations (1%) could enhance the nutritional composition, digestive and immune functions, and intestinal microbial community and metabolic profiles of juvenile sea cucumber. These findings provide a preliminary theoretical foundation for applying polysaccharides in aquaculture practices for sea cucumbers and other marine species.

## INTRODUCTION

The sea cucumber *Apostichopus japonicus* belongs to the genus *Apostichopus* within the class Holothuroidea of the phylum Echinodermata. As a globally recognized precious food material with dual medicinal and edible properties, it is rich in bioactive components such as polysaccharides, proteins and peptides, and saponins. It has significant anti-tumor activity and anti-coagulant properties, while can also regulate gut microbiota and enhance antioxidant enzyme activity. These biological functions contribute to improved immune response, anti-inflammatory effects, and tissue regeneration (*Lu et al., 2024*).

In artificial sea cucumber farming, feed quality is a critical factor limiting their growth. At present, the main ingredients of feed are macroalgae powder and animal/plant proteins in aquaculture. The macroalgaes commonly used in feed include *Sargassum thunbergii*, *Scagassum* and *Saccharina japonica*. However, macroalgaes usually contain non-starch polysaccharides (NSPs) such as cellulose, hemicellulose, and pectin which are resistant to degradation by endogenous animal enzymes. Moreover, their soluble non-starch polysaccharides (SNSPs) exhibit strong antinutritional effects. The digesta viscosity of SNSPs is increased when dissolved in water, which can impede nutrient absorption and alter gut microbiota composition, ultimately reducing feed digestibility in sea cucumbers (*Bedford & Classen, 1992*; *Wang, Xu & Zou, 2024*). To mitigate these effects, farmers supplement sea cucumber diets with sea urchin feces to counteract NSP/SNSP impacts (*Yu et al., 2023*). Additionally, researchers improved kelp-based feed quality using *Bacillus amyloliquefaciens* WB1 to degrade alginates in kelp. Pretreatment with this bacterial strain could significantly enhance digestive and immune enzyme activities in sea cucumbers (*Wang et al., 2015*). Therefore, developing novel macroalgae-based feeds remains crucial for sustainable aquaculture.

Studies have demonstrated that polysaccharides can form stable complexes with reactive oxygen species (ROS), enhancing immune function, and regulating gastrointestinal microbiota in animals. For example, it was showed that astragalus polysaccharide could improve immune and antioxidant enzyme activities, modulate intestinal non-specific immune responses, and alter microbial community structure of sea cucumber (*Song et al., 2019*; *Wang et al., 2024*), while could also increase resistance to

*Vibrio* infections (*Wang et al., 2009*). Furthermore, combination of β-glucan and mannooligosaccharides markedly boosted immune capacity of sea cucumbers (*Gu et al., 2011*). In addition, synergistic effects between microorganisms and polysaccharides have also been observed. For instance, co-administration of *Bacillus subtilis* and fructooligosaccharides substantially improved immunity and disease resistance in sea cucumbers (*Zhang et al., 2010*). These findings highlight the promising role of polysaccharides in sea cucumber aquaculture. With recent advancements in macroalgal polysaccharide extraction and purification techniques (*Dobrincic et al., 2021*; *Liu et al., 2020a*), research on algal-derived polysaccharides in breed is attracting increasing scientific interest (*Gu et al., 2011*; *Zhang et al., 2010*).

Brown algae polysaccharides (BAPs) primarily found in *Saccharina japonica*, *Sargassum fusiforme*, and *Undaria pinnatifida*, are classified into alginates, laminarin, and fucoidan and so on. They exhibit multifaceted biological functions including antioxidant, anti-inflammatory, immunomodulatory, antimicrobial, and antiviral activities (*Mazepa et al., 2022*). Since 2020, the addition of antibiotics in feed has been banned in China, and BAPs have emerged as promising bioactive molecules for developing eco-friendly antibiotic alternatives because of its no toxic side effects and no residue. It was reported that the growth performance, digestive enzyme activity, and immune parameters were significantly enhanced when feeding sea cucumbers with *Bacillus subtilis* and alginate oligosaccharides (AOS). Besides, they could also reduce pathogenic bacterial lethality, promote beneficial gut microbiota, and suppress harmful bacterial populations (*Wang et al., 2017*). Similarly, fucoidan from Okinawa Mozuku (*Cladosiphon okamuranus*) enhanced growth performance, feed efficiency, antioxidant capacity, and intestinal microbiota modulation feeding Zebrafish (*Ikeda-Ohtsubo et al., 2020*). Furthermore, fucoidan could also significantly improved the growth metrics, gut health, and antioxidant status in Nile tilapia (*Oreochromis niloticus*) (*Mahgoub et al., 2020*). Researchers also found that fucoidan could promote the growth, improve the digestive, immune and antioxidant capacity, increased the efficiency of lipid metabolism, and decreased the efficiency of glucose metabolism of *A. japonicus* (*Liu et al., 2025*). In addition, κ-selenocarrageenan could increase the antioxidant capacity and modulate the intestinal microbiota of sea cucumbers (*Wang et al., 2021*). In summary, these findings underscore the potential of BAPs as sustainable feed additives in aquaculture.

In this study, the BAP extracted from *Undaria pinnatifida* (UPS) with composition of alginate, mannitol, fucoidan and fucoheterosaccharides was studied to explore the combined effects of various BAPs. Different concentrations of UPS were supplemented to investigate its effects on growth performance, nutritional composition, immune enzyme activity, and intestinal microbiota. The optimal UPS dosage was determined, providing a scientific foundation for sustainable aquaculture practices of UPS.

## MATERIALS AND METHODS

### Experimental materials

The sea cucumber feed was purchased from Shandong Xuchang Biotechnology Co., Ltd. The compound feed additive "Microbial brick 85" (containing *Bacillus subtilis*,

Table 1 The main components of UPS.

| Components | Content |
|---|---|
| Sulphate based (%) | 8.46 |
| Alginic acid (%) | 25.1 |
| Mannitol (g/100 g) | 11.2 |
| Total sugar (as glucose, g/100 g) | 3.4 |
| Fucoidan (as fucose, g/100 g) | 13.2 |
| Fucoheterosaccharides (g/100 g) | 4.81 |
| Crude ash (g/100 g) | 58.9 |

*Lactobacillus acidophilus*, and *Saccharomyces cerevisiae*) was purchased from Shandong Ruilisheng Pharmaceutical Co., Ltd. The brown algae polysaccharide extracted from *Undaria pinnatifida* (UPS) was produced by Rongcheng Hongde Marine Biotechnology Co., Ltd., with detailed components listed in Table 1. The UPS has an ash content of 58.9%, and the remaining polysaccharide components are alginate, mannitol, fucoidan and fucoheterosaccharides.

## Feed fermentation and mixture preparation

Microbial Brick 85 (2% w/w) was added to the sea cucumber feed and fermented at 20 °C for 2 days. The fermented mixture was then blended with marine sediment at a ratio of 1:2.7 (feed:sediment) for feeding. To investigate the effects of UPS dosage on juvenile sea cucumber growth, six experimental groups were established with UPS supplementation levels of 0%, 0.5%, 1%, 1.5%, 2%, and 3% (polysaccharide/sea cucumber biomass), designated as HS0, HS1, HS2, HS3, HS4, and HS5, respectively.

## Acclimation and feeding experiment

Juvenile sea cucumbers were acclimated at a density of 40–50 individuals in 10 L filtered seawater for 6 days. Environmental conditions were maintained at 15 ± 1 °C, salinity of 31–33‰, pH 8.0, and dissolved oxygen levels of 10 mg/L. During the acclimation period, the animals were fed daily at 16:00 with a fermented feed mixture equivalent to 5% of their body weight. Residual feed and fecal matter were removed at 14:00 the following day using a plastic tubing siphon. Water exchange was performed as needed, with replacement volumes adjusted based on feeding activity and water quality parameters. As for the feeding experiment, a total of 6 kg of healthy, well-developed, symmetrical, and disease-free juvenile sea cucumbers were selected and were randomly distributed into 18 glass experimental tanks, with 0.33 kg per tank. There were three replicates in each experimental group. Each sea cucumber was blotted on dry filter paper for 30 s and weighed (recorded as W0) before being reintroduced into seawater. Growth and feeding conditions were the same as acclimation parameters.

## Sampling

Following a 60-day diet, the juvenile sea cucumbers were collected after 72 h of fasting. All sea cucumbers from each tank were collected and kept on dry filter paper for 30 s to

remove excess water. The sea cucumbers were weighed (WT), and the specific growth rate (SGR) was calculated. After weighing, 10 sea cucumbers were randomly selected from each tank and dissected on a chilled surface. Sterilized scissors were used to collect the coelomic fluid of 3 sea cucumbers in each tank and transferred into 1.5 mL centrifuge tubes for immunological function analysis. The intestinal tract of the other three sea cucumbers in each tank were placed into 1.5 mL centrifuge tubes for digestive enzyme activity assays. The intestinal tract of the remaining three sea cucumbers were quickly put into 1.5 mL centrifuge tubes and frozen in dry ice for 16S rRNA sequencing, and the body wall were used to determine the crude composition. The coelomic fluid, intestinal tract and body wall of the last 1 sea cucumber was collected and stored in −80 °C as the backup sample. The body wall weight (WB) and length (L) of the juvenile sea cucumbers were accurately measured. The intestinal tract was stretched to measure its weight (WI) and length (LI) after separation on a chilled surface. All samples were stored at −80 °C immediately after collection.

## Nutritional composition analysis

The chemical composition of the juvenile sea cucumber body wall was determined according to *AOAC (2005)* standard procedures. The samples were dried at 105 °C to determine the moisture content. Protein content in the intestine and coelomic fluid was analyzed using the Coomassie brilliant blue method (reagent purchased from Nanjing Jiancheng Bioengineering Institute, Nanjing, China), while protein content in the body wall was measured *via* the Kjeldahl method (JK9870 apparatus; Jinan Precision Analytical Instrument Co., Ltd., Jinan, China). Lipid content was determined by Soxhlet extraction. Polysaccharide content was quantified using the phenol-sulfuric acid method. Ash content was determined by incineration in a muffle furnace at 550 °C.

## Digestive enzyme activity and immunoenzyme assays

Intestinal digestive enzyme activities were measured using commercial kits: lipase (LPS, A054-1-1), pepsin (PP, A080-1-1), amylase (AMS, C016-1-1), and cellulase (CL, A138-1-1). Under ice bath conditions, 10% sea cucumber intestinal homogenate was prepared, centrifuged at 4,000 rpm for 10 min, and the supernatant was collected for use. The reaction system was prepared according to the kit operation manual, and the activity of each digestive enzyme was determined. For the determination of lipase activity, the absorbance was measured at 420 nm after the reaction system was prepared; for pepsin and amylase activity, the absorbance was measured at 660 nm after reaction solution was incubated at 37 °C for 20 min (pepsin) or 7.5 min (amylase); for cellulase activity, the reaction system was incubated at 37 °C for 30 min, and then immediately boiled for 15 min to terminate the reaction, the absorbance was measured at 550 nm to calculate cellulase activity. Coelomic fluid immune-related parameters, including superoxide dismutase (T-SOD, A001-1-2), total antioxidant capacity (T-AOC, A015-1-2), malondialdehyde (MDA, A003-1-1), and hydrogen peroxide ($H_2O_2$, A064-1-1), were analyzed with corresponding assay kits. For the determination of T-SOD activity, the reaction solution was prepared according to the kit instructions, incubated at 37 °C for 40 min, and the

absorbance was measured at 550 nm; for T-AOC capacity, the reaction solution reacted at 37 °C for 30 min and measured at 520 nm; for the determination of MDA content, the reaction solution was mixed evenly, incubated at 95 °C for 40 min, centrifuged at 4,000 rpm for 10 min, and the supernatant was taken and the absorbance was measured at 532 nm; for the measurement of hydrogen peroxide content, the absorbance was determined at 405 nm. All kits were purchased from Nanjing Jiancheng Bioengineering Institute.

## 16S rRNA sequencing and bioinformation analysis

The intestinal microbial DNA of juvenile sea cucumbers was extracted using the MagPure Soil DNA LQ Kit (Magen), followed by amplification of the 16S rRNA gene V3–V4 region with primers 343F and 798R. The PCR products were purified with AMPure XP beads and sequenced on the Illumina NovaSeq 6000 platform to generate 250-bp paired-end reads. Preprocessing of paired-end reads was conducted using Cutadapt software to identify and remove adapter sequences. The reads underwent quality filtering, denoising, merging, and chimeric read detection and removal with DADA2 with the default parameters of QIIME2, resulting in representative reads and ASV abundance data. Subsequently, all representative reads were annotated and blasted against the Silva database (Version 138), utilizing q2-feature-classifier with its default parameters. Sequencing services were provided by Shanghai Ouyi Biotechnology Co., Ltd. (Shanghai, China).

ASV flower plot was performed using the OECloud tools at https://cloud.oebiotech.com.

The data was normalized using the following steps. Firstly, the data were converted to proportions by dividing the reads of each ASV in the sample by the total reads in the sample, and analyzed using relative abundance. Then the data were pooled by diluting the data by randomly resampling each sample to the minimum sequencing depth of the sample. Alpha and beta diversity analyses were conducted in QIIME 2. Alpha diversity indices (Chao1, ACE, Good's coverage, and Simpson) were calculated. One-way analysis of variance (ANOVA) was applied to evaluate the significant differences in the bacterial α-diversity indices between the communities. Principle coordinate analysis (PCoA) by R package was used to determine the differences in the bacterial community structures based on Bray-Curtis distances. Permutational multivariate analysis of variance (PERMANOVA) was performed using the Adonis function to evaluate the effects of UPS supplementation on the variations in the bacterial communities. Significant analysis of differential microbial species at family and genus levels were tested using Kruskal Wallis and $P < 0.05$ was considered statistically significant. Linear discriminant analysis effect size (LEfSe) analysis was used to determine the microbial taxa that best characterized each study group. Linear discriminant analysis (LDA) scores $> 2.0$ and $P < 0.05$ were considered significantly. Functional profiling of the intestinal microbiota was predicted with PICRUSt2 (*Yan et al., 2023*). Differential metabolic pathways were analyzed using the Kruskal-Wallis test.

Pearson correlation analysis was applied to evaluate associations between microbial taxa, biochemical indicators, and intestinal microbial communities. $0.1 < |\text{Pearson-}r| < 0.3$ was considered as weak correlation; $0.3 < |\text{Pearson-}r| < 0.5$ was medium correlation and significant correlations were defined as $|\text{Pearson-}r| > 0.5$ with $P < 0.05$.

## Statistical analysis

SPSS 22.0 (SPSS Inc., Chicago, IL, USA) was used to analyze the test of their normality and homoscedasticity. All results were presented as mean ± S.E.M ($n$ = 3 or 5). One-way analysis of variance (ANOVA) was used for multi-group comparison, and Tukey test was used. $P < 0.05$ was considered statistically significant. Growth data were calculated, including the weight gain rate (WGR% = (WT − W0)/W0 × 100), specific growth rate (SGR% = (LnWT − LnW0)/number of days × 100) (*Li et al., 2024*), the intestinal body wall weight ratio (R = WI/WB × 100) and the intestinal wall length ratio (RGL = LI/L × 100). Data visualization processing was performed using Origin 2024.

## RESULTS

### Effects of UPS on the growth index

After 60 days of feeding, there was no significant difference in the weight gain rate WGR, SGR, R and RGL (Table S1) indicating that UPS have few effects on the growth of juvenile sea cucumbers.

### Effects of UPS on the nutritional composition

Analysis of the nutritional components in juvenile sea cucumbers revealed that the polysaccharide content of body wall in HS2 group exhibited approximately 1.6-fold higher than that in HS0 group ($P < 0.05$). However, UPS supplementation had no significant effect on protein, lipid, ash, or moisture content in the body wall ($P > 0.05$) (Table 2). These results suggested that low UPS supplementation (at 1% concentration) could enhance polysaccharide accumulation in juvenile sea cucumbers, thus improving quality of sea cucumbers.

### Effects of UPS on the intestinal digestive enzyme activity

The activities of cellulase, lipase, pepsin, and amylase of the intestinal tract of juvenile sea cucumbers were measured. As demonstrated in Table 3, the HS2 group exhibited significantly higher cellulase activity than HS0 and other experimental groups ($P < 0.05$), which was 5.8-fold higher than that of HS0 (Table 3). Additionally, lipase activity in all UPS-supplemented groups exceeded that of HS0, with HS5 group showing 1.6-fold higher than HS0 group. However, no significant differences in pepsin or amylase activities were observed between experimental and control groups (Table 3). These results showed that low UPS supplementation (1%) could increase the cellulase activity and UPS gradually increased the lipase activity of sea cucumbers with the increase of concentration.

### Effects of UPS on the immune performance

To investigate the effects of UPS on immune performance in juvenile sea cucumbers, the activity of superoxide dismutase (T-SOD), total antioxidant capacity (T-AOC), and the contents of malondialdehyde (MDA) and hydrogen peroxide ($H_2O_2$) in coelomic fluid were analyzed. As shown in Table 4, T-AOC kept unchnaged with the increase of UPS concentration. The T-SOD activities in all UPS-supplemented groups were significantly higher than in HS0 ($P < 0.05$), with the HS2 and HS3 groups showing the most

**Table 2 Effects of UPS on nutritional composition of juvenile sea cucumbers.**

| | UPS concentration (groups) | | | | | |
|---|---|---|---|---|---|---|
| | HS0 | HS1 | HS2 | HS3 | HS4 | HS5 |
| Polysaccharide content (% dry weight) | 2.78 ± 0.49[b] | 2.76 ± 0.3[b] | 4.46 ± 0.49[a] | 3.01 ± 0.47[b] | 2.81 ± 0.16[b] | 2.69 ± 0.66[b] |
| Protein content (% wet weight) | 3.94 ± 0.36[a] | 4.40 ± 0.22[a] | 4.61 ± 0.17[a] | 4.99 ± 0.38[a] | 4.15 ± 0.44[a] | 4.32 ± 0.61[a] |
| Lipid content (% wet weight) | 2.43 ± 0.15[a] | 2.55 ± 0.13[a] | 2.51 ± 0.07[a] | 2.45 ± 0.05[a] | 2.07 ± 0.16[a] | 2.33 ± 0.13[a] |
| Ash (% wet weight) | 3.65 ± 0.38[ab] | 4.40 ± 0.04[a] | 3.97 ± 0.41[ab] | 3.12 ± 0.13[b] | 3.50 ± 0.11[ab] | 3.11 ± 0.97[b] |
| Moisture (% wet weight) | 89.13 ± 0.30[a] | 87.60 ± 0.62[a] | 89.49 ± 0.31[a] | 87.56 ± 1.51[a] | 90.28 ± 0.45[a] | 89.96 ± 0.54[a] |

Note:
Data are expressed as means ± S.E.M ($n$ = 3). Different letters show a significant difference ($P < 0.05$).

**Table 3 Effects of UPS on the intestinal digestive enzyme activity of juvenile sea cucumbers.**

| | UPS concentration (groups) | | | | | |
|---|---|---|---|---|---|---|
| | HS0 | HS1 | HS2 | HS3 | HS4 | HS5 |
| Cellulase (U/mg prot) | 2.50 ± 0.48[b] | 4.46 ± 1.18[b] | 14.53 ± 1.00[a] | 4.24 ± 0.33[b] | 4.11 ± 0.89[b] | 5.19 ± 0.89[b] |
| Lipase (U/mg prot) | 48.36 ± 7.62[b] | 58.26 ± 7.96[ab] | 64.14 ± 1.75[ab] | 56.69 ± 6.68[ab] | 73.36 ± 5.00[ab] | 77.68 ± 4.84[a] |
| Pepsin (U/mg prot) | 1.56 ± 0.45[a] | 1.09 ± 0.18[a] | 1.25 ± 0.25[a] | 0.79 ± 0.08[a] | 1.30 ± 0.28[a] | 0.96 ± 0.11[a] |
| Amylase (U/mg prot) | 1.01 ± 0.29[a] | 1.01 ± 0.04[a] | 1.13 ± 0.30[a] | 1.23 ± 0.20[a] | 1.28 ± 0.09[a] | 1.25 ± 0.02[a] |

Note:
Data are expressed as means ± S.E.M ($n$ = 3). Different letters show a significant difference ($P < 0.05$).

**Table 4 Effects of UPS on immune performance of juvenile sea cucumbers.**

| | UPS concentration (groups) | | | | | |
|---|---|---|---|---|---|---|
| | HS0 | HS1 | HS2 | HS3 | HS4 | HS5 |
| T-SOD (U/mL) | 238.58 ± 6.32[c] | 280.9 ± 8.58[bc] | 337.82 ± 15.53[a] | 336.53 ± 10.02[a] | 313.63 ± 8.93[ab] | 278.49 ± 13.54[bc] |
| T-AOC (U/mL) | 2.61 ± 0.3[a] | 2.87 ± 0.18[a] | 2.82 ± 0.22[a] | 3.65 ± 0.3[a] | 3.63 ± 0.1[a] | 3.69 ± 0.44[a] |
| H2O2 (U/mL) | 37.24 ± 0.93[d] | 54.26 ± 3.44[b] | 64.73 ± 3.09[a] | 51.56 ± 1.97[bc] | 52.35 ± 3.23[bc] | 42.06 ± 1.42[cd] |
| MDA (U/mL) | 1.16 ± 0.26[bc] | 1.11 ± 0.06[bc] | 0.77 ± 0.11[cd] | 0.43 ± 0.11[d] | 1.87 ± 0.13[a] | 1.57 ± 0.14[ab] |

Note:
Data are expressed as means ± S.E.M ($n$ = 3). Different letters show a significant difference ($P < 0.05$).

pronounced increase (1.4-fold higher than HS0, $P < 0.05$). Conversely, MDA content in low-UPS supplementation group (HS3) was significantly lower than that in HS0, whereas in high-UPS groups (HS4 and HS5) was higher. These results indicated that UPS supplementation especially with low UPS concentration could obviously improve the immune performance of juvenile sea cucumbers. However, the content of $H_2O_2$ in all UPS supplementation groups were higher than HS0 group indicating that UPS supplementation could also lead to oxidative stress in some degree.

## Effects of UPS on the intestinal microbiota diversity

As shown in Fig. 1A, no significant difference in ASV counts was observed between the control group (HS0) and UPS-supplemented groups ($P > 0.05$) (Fig. 1A). The rarefaction curves tend to flatten as the number of extraction sequences increases indicating that the

A

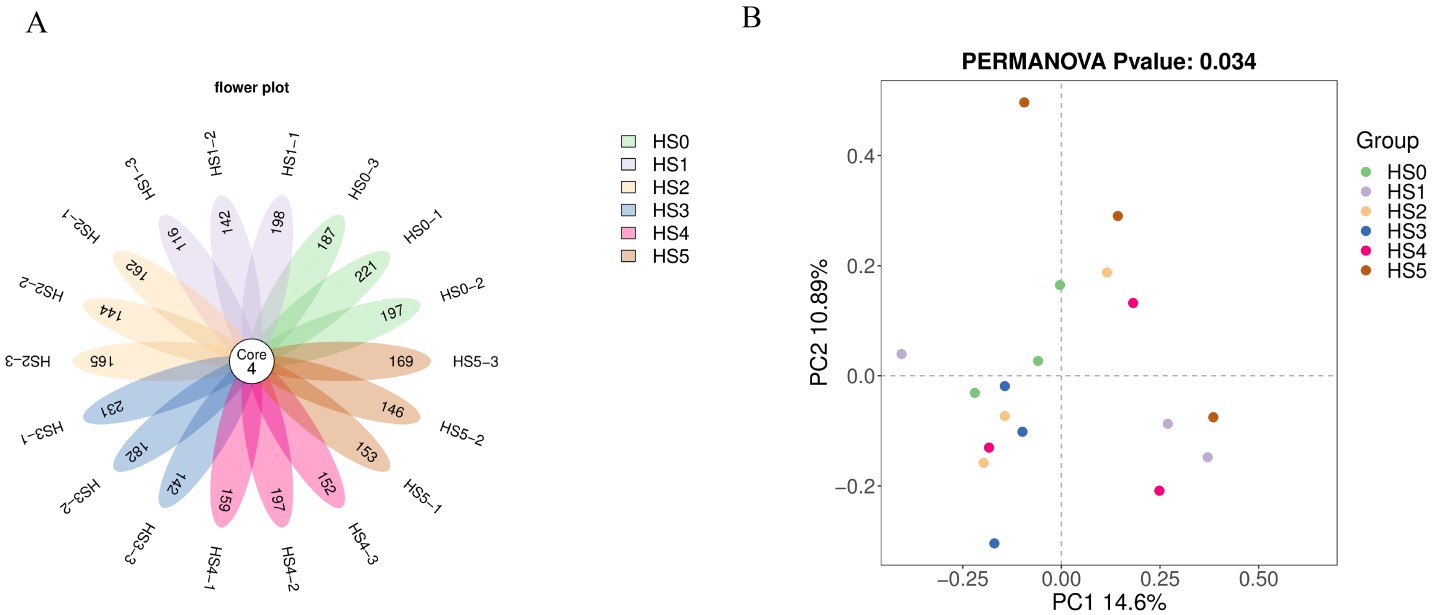

B

**Figure 1 Flower plot of ASVs (A) and PCoA analysis of community between different UPS treatments (B).**

amount of sequencing data of samples was reasonable (Fig. S1). The β-diversity was analyzed using Bray-Curtis distance. The intestinal bacterial communities (PERMANOVA, R = 0.692 and $P = 0.034 < 0.05$) were significantly differentiated based on UPS supplementation (Fig. 1B), indicating that UPS supplementation could affected the intestinal bacterial communities of juvenile sea cucumbers.

Alpha diversity across groups was assessed using the ACE index, Chao1 index, Good's coverage, and Simpson index. As demonstrated in Fig. 2, both ACE and Chao1 indices in UPS-supplemented groups showed non-significant decreases compared to HS0. Although the Simpson index significantly declined in the HS5 group ($P < 0.05$), UPS supplementation exhibited no significant impact on the alpha diversity of intestinal microbiota in juvenile sea cucumbers in other groups (Fig. 2).

## Analysis of intestinal microbial composition

"Microbial brick 85" (containing *Bacillus subtilis*, *Lactobacillus acidophilus*, and *Saccharomyces cerevisiae*) was used to ferment the feed of sea cucumber. In order to detect its effect of intestinal bacterial communities, the intestinal microorganisms of sea cucumber were cultured. The result revealed that only one strain of *Bacillus subtilis* was obtained from each of the HS1, HS2, and HS4 groups in MRS medium, while no *Lactobacillus acidophilus* was isolated (Table S2). In addition, it was found that these three strains had poor salt tolerance and could not be obtained on the seawater culture medium, indicating that the microbial strains of the brick could not survive in the sea cucumber breeding environment, so the influence on the intestinal microbial community of the sea cucumber could be ignored.

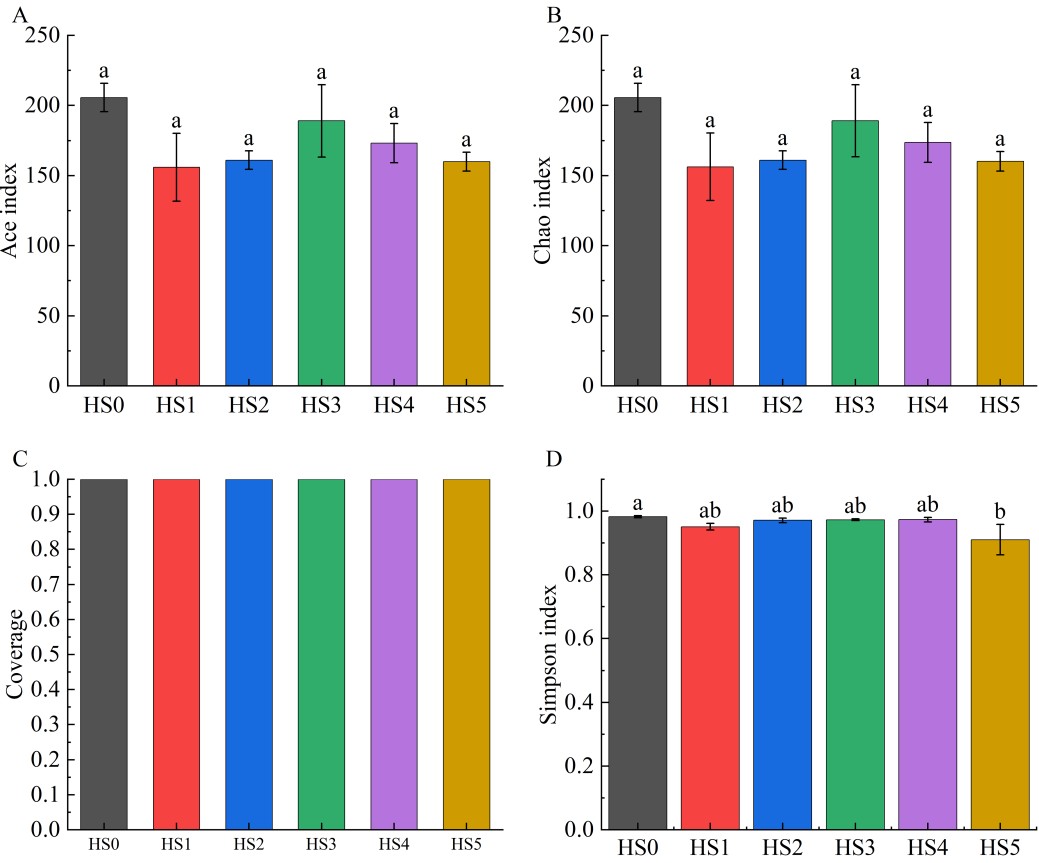

**Figure 2 Alpha diversity of intestinal microbiota in sea cucumbers.** Community diversity index including Ace index (A), Chao 1 index (B), Good's coverage (C), and Simpson index (D). Different lowercase letters show a significant difference ($P < 0.05$).

The intestinal microbiota composition of juvenile sea cucumbers at the phylum level is shown in Fig. 3A. In HS0 and UPS-supplemented groups, the dominant phyla were primarily Bacteroidota, Proteobacteria, Firmicutes, and Actinobacteriota. No significant differences in microbial abundance were observed among these groups (Fig. 3A). At the family level, families with an average relative abundance > 1% were analyzed. As shown in Fig. 3B, the dominant families in all groups included Muribaculaceae, Rhodobacteraceae, Prevotellaceae, Vibrionaceae, and Lachnospiraceae. The highest relative abundance of Muribaculaceae (24.1%) was observed in HS0, while all UPS-supplemented groups exhibited lower values, with the lowest in HS5 (12.5%). The abundance of Rhodobacteraceae was higher in all groups compared to HS0 except HS4. No significant differences in Prevotellaceae abundance were detected between the control and UPS-supplemented groups. Vibrionaceae abundance varied markedly among groups, ranging from 0.9% in HS2 to 18.9% in HS1 (Fig. 3B). At the genus level, the dominant genera across groups were *Muribaculaceae*, *Prevotellaceae* UCG-001, *Vibrio*, and *Alistipes* (Fig. 3C).

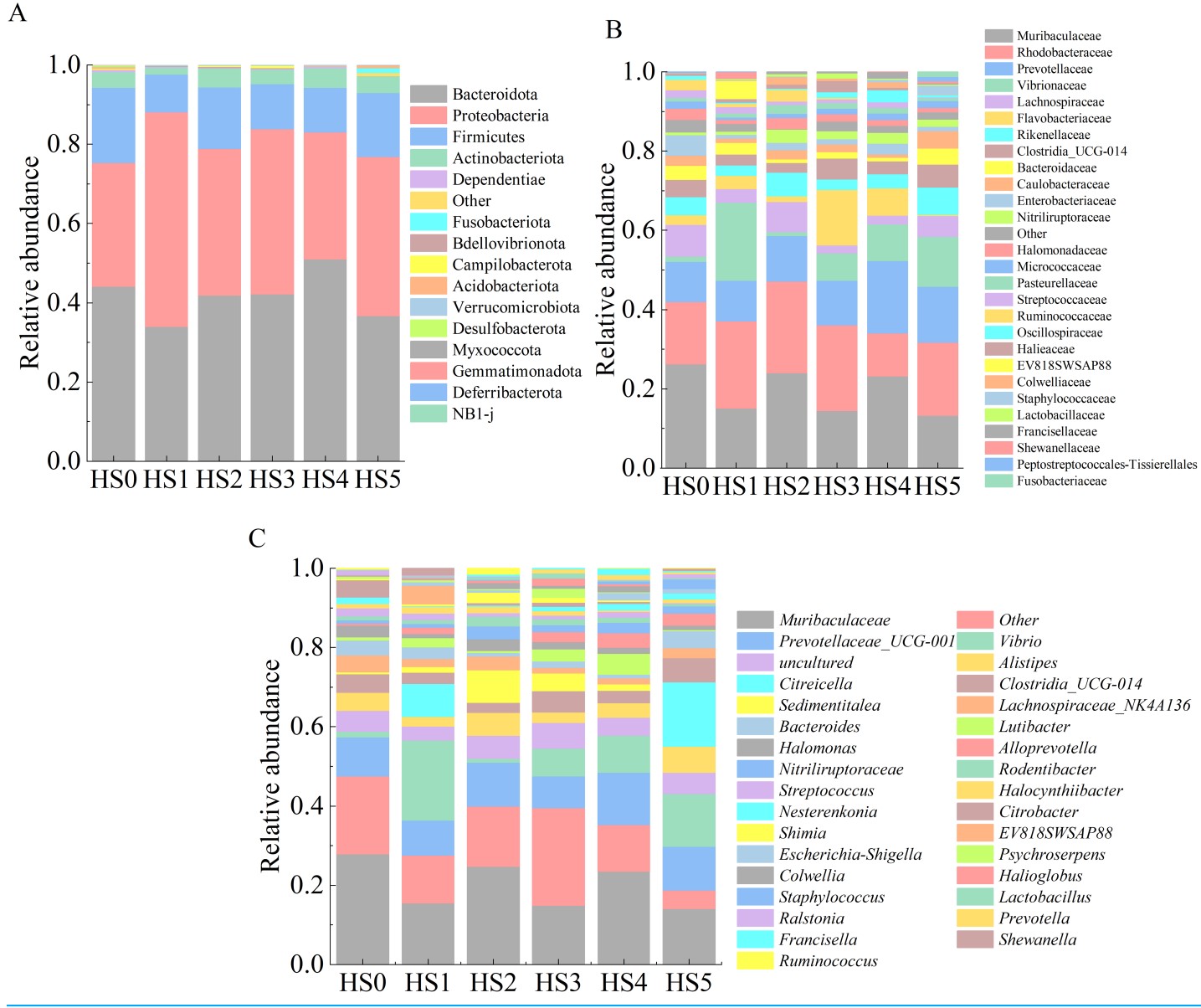

**Figure 3 Composition of intestinal microbiota in juvenile sea cucumbers.** Phylum (A); family (B); genus (C).

Differential microbial species among groups were analyzed at the family and genus levels. As shown in Fig. 4A, the differential families were EV818SWSAP88, Burkholderiaceae, Peptostreptococcales-Tissierellales, unknown family and BD2-7 ($P < 0.05$). It is worth noting that the abundance of Burkholderiaceae family strains was correlated with the amount of UPS added. In low UPS supplementation groups (HS1, HS2 and HS3), the abundance of Burkholderiaceae family strains decreased with the improvement of UPS concentration while the high-concentration addition group showed an opposite trend (Fig. 4A). At the genus level, Genera with significant intergroup

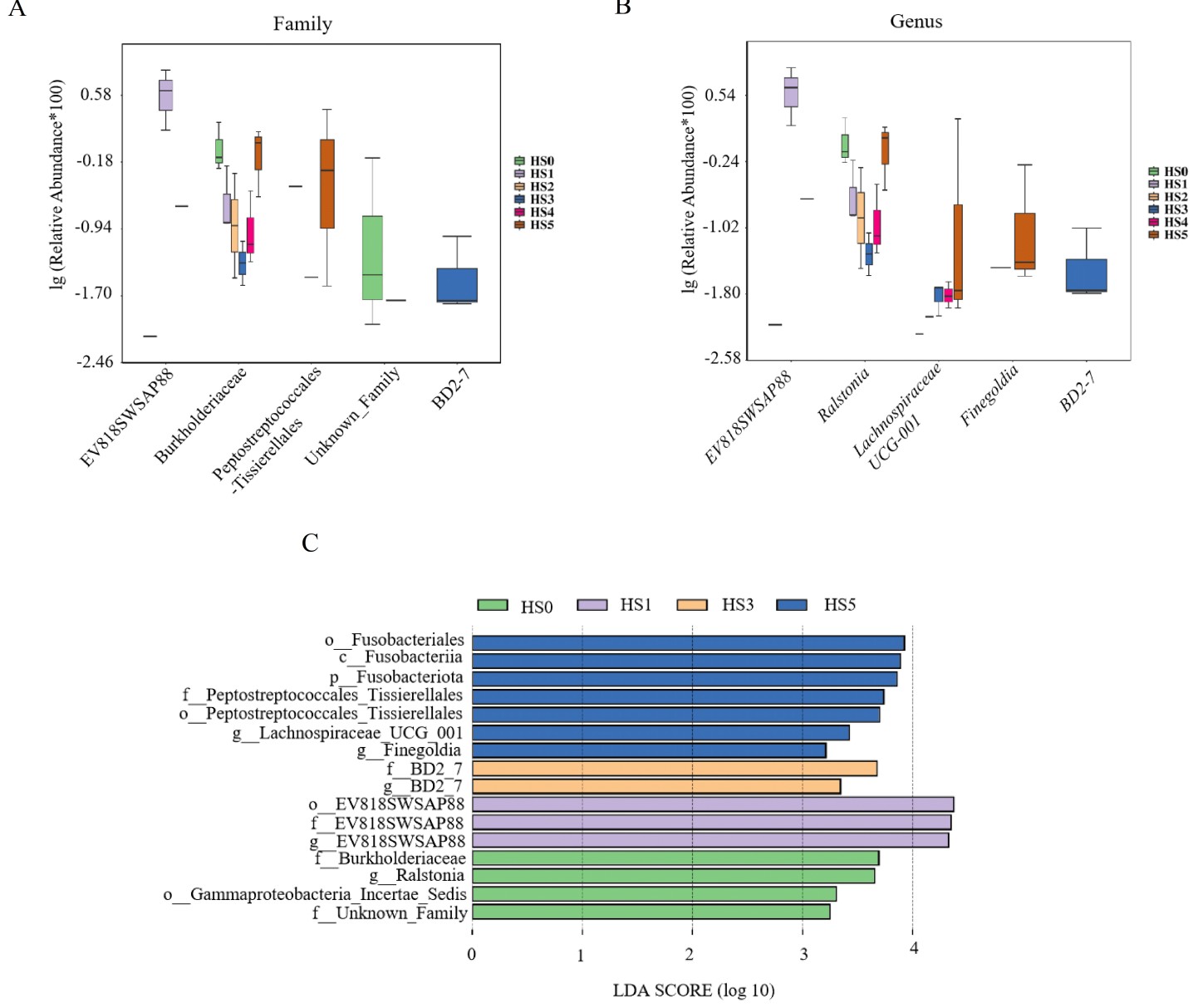

**Figure 4 Differential microbial species among different groups.** Family level (A); genus level (B); LEfSe analysis (C).

differences in relative abundance included *EV818SWSAP88*, *Ralstonia*, *Lachnospiraceae* UCG-001, *Finegoldia*, and *BD2-7* (Fig. 4B). The abundance of *Ralstonia* strains showed the same trend with the Burkholderiaceae family strains , and their abundance was closely related to UPS concentration. Morever, the abundance of *Lachnospiraceae* UCG-001 strains increased in HS3, HS4 and HS5 groups. The result of LEfSe analysis was consistent with Figs. 4A, 4B showing the high abundance of Burkholderiaceae family and *Ralstonia* strains in HS0 (Fig. 4C). These results demonstrated that low UPS supplementation could

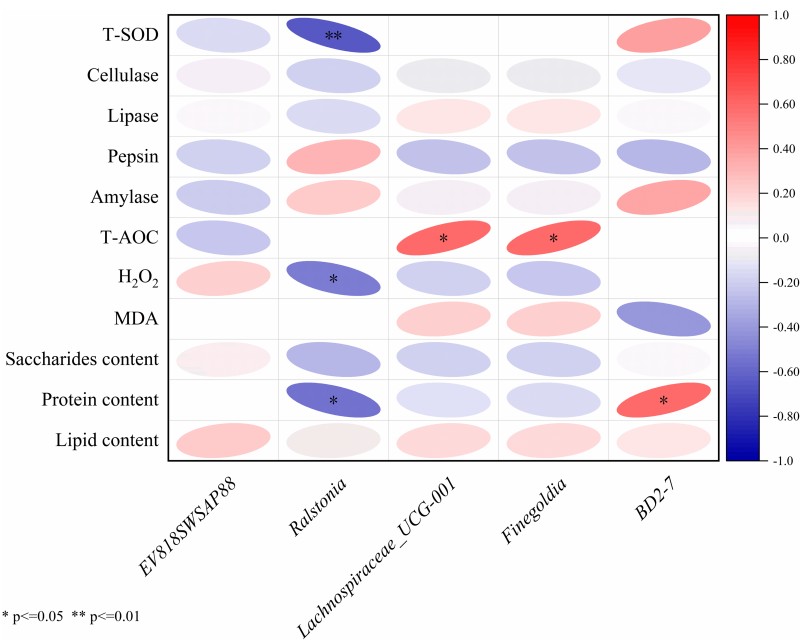

* p<=0.05  ** p<=0.01

**Figure 5 Correlation analysis between intestinal microbiota and with nutritional composition, immunity index and digestive enzyme activities.**

significantly reduce the relative abundance of the pathogenic microorganisms (*Ralstonia*) and change the microbial structure of juvenile sea cucumbers.

## Functional analysis of intestinal microbiota

Functional prediction analysis using PICRUSt2 revealed microbial functional profiles at three hierarchical levels. At level 1, the control group (HS0) and experimental groups shared six functional categories: cellular processes, environmental information processing, genetic information processing, human diseases, metabolism, and organismic systems. Among these, metabolism was the predominant category, though no significant intergroup differences in functional abundance were observed (Fig. S2A). At level 2, key functional pathways included amino acid metabolism, carbohydrate metabolism, cellular community, and energy metabolism. While minor variations occurred in cell motility and signal transduction, there were no significant differences in other functional pathways (Fig. S2B). Further functional analysis was conducted at the level 3, showing mainly some differences in the three metabolic pathways such as nitrotoluene degradation, nonribosomal peptide structures, and pentose and glucuronate interconversions.

 With increasing UPS concentration (excluding HS1), these pathways exhibited progressive enhancement. The HS5 group reached the highest level. However, no significant differences were detected between HS2, HS3, HS4, and the control (Fig. S2C).

## Correlation analysis

The results of Pearson correlation analysis of five significantly divergent microorganisms with nutritional composition, immunity index and digestive enzyme activities were shown

in Fig. 5. The *Ralstonia* strain had a significant negative correlation with the $H_2O_2$ content ($r = -0.51$, $P < 0.05$), the protein content ($r = -0.55$, $P < 0.05$) and the T-SOD enzyme activity ($r = -0.65$, $P < 0.01$). *Lachnospiraceae* UCG-001 ($r = 0.59$, $P = 0.01$) and *Finegoldia* ($r = 0.59$, $P = 0.01$) had a significant positive correlation with T-AOC capacity, and *BD*2-7 significantly correlated with protein content ($r = 0.59$, $P = 0.01$).

## DISCUSSION

The *Undaria pinnatifida*-derived BAP (UPS) was a mixture of carbohydrates including alginate, mannitol, fucoidan and fucoheterosaccharides and so on (Table 1). This study aimed to research the combined effects of various BAPs on the nutritional composition, immunity, digestion and intestinal microbiota of juvenile sea cucumber. As shown in Table S1, UPS supplementation had no significant effect on the growth of juvenile sea cucumber, which was consistent with the feeding effect of astragalus polysaccharides (*Song et al., 2019*). While the growth performance of Nile tilapia and piglets was significantly promoted when feeding them with fucoidans or seaweed-derived laminarin and fucoidan, respectively (*Heim et al., 2014*; *Mahgoub et al., 2020*). These results indicated that the components or the ratio of polysaccharides in BAPs play a significant role in their function. Analysis of the nutritional components of the sea cucumber body wall revealed that 1% UPS supplementation significantly increased the polysaccharide content, which was 1.6-fold higher than that of the control group, indicating that low UPS supplementation can improve the nutritional components of the sea cucumber body wall and enhance the quality of sea cucumber (Table 2). Similarly, the previous study demonstrated the body wall crude fat and polysaccharide contents of *Apostichopus japonicus* reached their maximum with low fucoidan supplementation (1.41%), which was 1.17-fold higher than the control group (*Liu et al., 2025*). While when the UPS concentration was high (2% and 3%), the polysaccharide content of sea cucumber body wall did not increase significantly, indicating that the excessive addition of UPS may have exceeded the body's ability to process it, and the reducing sugar content exceeded the body's tolerance limit, affecting the intestinal absorption of nutrients (*Wang et al., 2023*). These results showed that the low concentration of BAP had a positive effect on the nutritional quality of the animals.

Analysis of digestive enzyme activities in the intestinal tract of sea cucumbers revealed that UPS supplementation could elevate the cellulase and lipase activities. The 1% UPS-supplemented group displayed a significant increase in intestinal cellulase activity (Table 3), which was 5.8-fold higher than the control group, correlating with elevated polysaccharide content in the body wall (Table 2). This result indicated that UPS supplementation could induce the production of cellulases and thus accelerated the decomposition of UPS into small molecule polysaccharide, and improved the utilization rate of UPS. Moreover, lipase activity exhibited a dose-dependent increase with rising UPS concentrations ($P < 0.05$), which was 1.6-fold higher than the control group, indicating the enhanced digestive capacity following UPS supplementation. These results were consistent with prior research demonstrating fucoidan's role in promoting digestive tract maturation and enhancing lipase activity (1.64 fold) in large yellow croaker (*Yin et al., 2022*),

indicating that the BAPs enhance digestive capacity by improving relevant digestive enzyme activities. However, the lipid content of the body remain unchanged may because of insufficient supply of lipid substances in the feed and lipid and UPS co-supply may be considered in later feeding.

The oxidative stress reaction usually occurs when the breeding environment or nutrient supply are unbalanced leading to the excessive production and accumulation of reactive oxygen species (ROS) in sea cucumbers, and the antioxidant system of the body is activated. It will lead to sea cucumber disease and even death if the oxidative stress cannot be controlled in time (*Tan, Norhaizan & Liew, 2018*). Superoxide dismutase (SOD) in living organisms is the first antioxidant line, removing excess ROS to generate hydrogen peroxide ($H_2O_2$) to reduce cellular oxidative damage (*Valavanidis et al., 2006*). The T-SOD activity in the UPS-supplementation group were improved, which increased 1.41 fold than the control group, indicating the positive role of UPS in activating the immune performance in juvenile sea cucumber (Table 4). This result was consistent with feeding sea cucumber with enzymatically hydrolyzed kelp powder (EKP, 1.18-fold higher) (*Wang et al., 2023*), and β-glucan and mannan oligosaccharide (1.4-fold higher) (*Gu et al., 2011*). The content of lipid peroxidation product malondialdehyde (MDA), which can reflect the rate and intensity of lipid peroxidation in the body and the degree of tissue peroxidation damage (*Atli & Canli, 2007*), was researched. It was found that the MDA content of the low-UPS supplemention group (1% and 1.5%) decreased while it was increased in the higher-UPS supplemention group (2% and 3%) compared with that in the control group (HS0), indicating that low UPS supplemention could reduce lipid peroxidation, whereas high concentrations had the opposite effect. The reason for this result may be due to the nutritional imbalance of high UPS concentration in the aquaculture environment and the deterioration of water quality, leading to the enhancement of lipid peroxidation. Similarly, it was found that lipid oxidation in pork was reduced and the quality of pork was improved when feeding pigs with laminarin and fucoidan polysaccharides from *Laminaria digitata* (*Moroney et al., 2015*). In conclusion, the UPS supplement can improve the immune performance of juvenile sea cucumber, which has the same effect as other polysaccharides such as astragalus polysaccharide, β-glucan, mannan oligosaccharide and fructooligosaccharide (*Gu et al., 2011*; *Wang et al., 2009*; *Zhang et al., 2010*). However, high $H_2O_2$ content was also detected in UPS supplementation groups which indicating the oxidative stress in sea cucumber. Furthermore, the total antioxidant capacity T-AOC which can reflect the overall oxidative stress level and reflect the antioxidant capacity in the body kept unchanged in the UPS-supplementation group (*Cao et al., 2010*). The T-AOC capacity contain enzymatic and non-enzymatic antioxidation system. The enzymatic antioxidation system contains superoxide dismutase (SOD), glutathione peroxidase (GSH-PX) and catalase (CAT), glutathione S transferase (GST), *etc*. The increase of T-SOD activity led to the production of $H_2O_2$, while CAT did not remove hydrogen peroxide in time, resulting in excessive $H_2O_2$ content. This may be the reseason of no significant increase of growth index with the UPS supplementation. Microencapsulation technique could control the release of embedded material and was a simple, safe and reliable method (*Vidhya Hindu et al., 2018*; *Sun et al., 2024*). In the future study, we will try to make UPS

microcapsules to sustainedly release polysaccharides into the sea cucumber growth environment and avoid oxidative stress caused by one-time addition.

The intestinal microbiota has a complex relationship with their hosts, and its abundance and composition can affect various physiological activities of the body, such as nutrient metabolism, nutrient balance, immune function, and development (*Liu et al., 2020b*, *2020c*; *Kang et al., 2023*). Previous studies have shown that intestinal microbiota had a significant effect on the growth of *Anguilla marmorata* and differences in intestinal microbial composition affect the growth of sea cucumber (*Yamazaki et al., 2016*). In this study, the intestinal microbiota of juvenile sea cucumber with UPS supplementation was researched. The result indicated that UPS supplementation had no significant effect on the abundance and diversity of intestinal microorganisms according to the α-diversity analysis (Fig. 2), while the control group and UPS-supplementation groups displayed distinct clustering showing by the β-Diversity analysis (Fig. 1B). Analysis of different species of biological community showed that UPS could decrease the abundance of Burkholderiaceae family strains and *Ralstonia* strains in low UPS supplementation groups (0.5–1.5%) while increase the abundance of them at high concentrations (2% and 3%) (Figs. 4A, 4B). *Ralstonia* belongs to Burkholderiaceae family, and the *Ralstonia* genus species are common opportunistic pathogens in water and soil (*Ryan & Adley, 2014*). Previous studies indicated that BAPs regulated intestinal microbiota through two mechanisms: on the one hand, it could promote the growth of beneficial bacteria to generate health-associated metabolites; on the other hand, it could inhibit intestinal pathogens or opportunistic microorganisms to reduce harmful metabolites (*Cires et al., 2019*). These results indicated that low UPS supplementation could protect sea cucumbers from pathogenic bacteria. Additionally, BAP fermentation could increase total organic acids in the intestine, lowering intestinal pH, which further suppressed harmful bacterial metabolites and contributed to intestinal homeostasis (*Nakata et al., 2016*). For example, it was demonstrated that β-glucan could promote Erythrobacteraceae proliferation, reduce Flavobacteriaceae abundance, and regulate microbial balance, and thereby enhanced intestinal immune responses in sea cucumber (*Yang et al., 2015*). The fucoidan could increase the abundance of *Lactobacillus* and Ruminococcaceae to regulate the gut microbiota in mice (*Shang et al., 2016*). Combination of fucoidan and galactooligosaccharides supported *Lactobacillus casei* growth and improved the high-fat diet-induced dyslipidemia (*Chen et al., 2019*). In summary, BAPs regulate intestinal microbiota by stimulating probiotics and inhibiting pathogenic bacteria.

The intestinal microbial function analysis showed that there was no significant difference in functional profiles at Level 1 and Level 2, while in Level 3 functions some changes in nonribosomal peptides structures, nitrotoluene degradation, and pentose and glucuronate interconversions in UPS supplementation groups (Fig. S2). Microorganisms could use potential carbon sources such as nitrotoluene to ensure their energy sources by nitrotoluene degradation (*Ju & Parales, 2011*). In addition, the addition of UPS could enhance pentose and glucuronate interconversion, improving the efficiency of sugar

conversion, and promoting the absorption and utilization of sugar. The changes in these metabolic pathways will be confirmed by metagenomic sequencing and qRT-PCR methods in the later research.

The Pearson correlation analysis revealed that the abundances of the pathogenic bacterium *Ralstonia* exhibited negative correlations with total superoxide dismutase (T-SOD) activity, $H_2O_2$ and protein content. UPS supplementation reduced the relative abundance of *Ralstonia*, thereby improving the immune performance of sea cucumbers. These findings further demonstrated that UPS could restructure the intestinal microbiota of sea cucumbers, synergistically enhancing their digestive and immune functions, ultimately elevating their nutritional value.

## CONCLUSIONS

This study investigated the effects of BAP derived from *Undaria pinnatifida* (UPS) on the nutritional composition, digestive capacity, immune performance and intestinal microbiota of juvenile sea cucumber (*Apostichopus japonicus*). The result demonstrated that UPS, particularly at 1% supplementation, significantly increased the polysaccharide content in the body wall, enhanced cellulase and lipase activities in the intestine, and improved the utilization efficiency of UPS. Furthermore, UPS elevated antioxidant capacity and immune performance by enhancing T-SOD enzyme activity and reducing lipid oxidation in sea cucumbers. Intestinal microbiota analysis revealed that low UPS supplementation suppressed *Ralstonia* abundance, thereby protecting pathogenic microorganism from sea cucumber. These results preliminarily consider that UPS as a high-quality feed additive for sea cucumbers. Future research should increase the sampling number or carry out large-scale breeding experiments to verify the effects of UPS and elucidate the mechanisms underlying its digestive and immune-enhancing effects, make UPS into microcapsules for controlled release and explore its applications in other aquaculture species.

## ACKNOWLEDGEMENTS

We thank Rongcheng Hongde Marine Biotechnology Co., Ltd. for providing the *Undaria pinnatifida*-derived Brown algae polysaccharide. We are grateful to the reviewers whose comments and suggestions greatly improved our manuscript.

### Funding

This work was supported by the Shandong Mariculture Innovation and Entrepreneurship Community Project (YZ2025001); National Undergraduate Training Program for Innovation and Entrepreneurship (202410214047). The funders had no role in study design, data collection and analysis, decision to publish, or preparation of the manuscript.

## Grant Disclosures

The following grant information was disclosed by the authors:

Shandong Mariculture Innovation and Entrepreneurship Community: YZ2025001.
National Undergraduate Training Program for Innovation and Entrepreneurship: 202410214047.

## Competing Interests

Wei Zhou is employed by Rongcheng Hongde Marine Biotechnology Co., Ltd. Yongjun Sun and Wenming Ju are employed by Homey Group Co., Ltd. The authors declare there are no competing interests.

## Author Contributions

- Jinghao Dong conceived and designed the experiments, authored or reviewed drafts of the article, and approved the final draft.
- Hailong Ma performed the experiments, authored or reviewed drafts of the article, and approved the final draft.
- Kuimei Liu analyzed the data, prepared figures and/or tables, and approved the final draft.
- Wei Zhou performed the experiments, authored or reviewed drafts of the article, and approved the final draft.
- Suya Liu performed the experiments, authored or reviewed drafts of the article, and approved the final draft.
- Yongjun Sun performed the experiments, authored or reviewed drafts of the article, and approved the final draft.
- Wenming Ju performed the experiments, authored or reviewed drafts of the article, and approved the final draft.
- Shanshan Wang conceived and designed the experiments, analyzed the data, prepared figures and/or tables, and approved the final draft.

## DNA Deposition

The following information was supplied regarding the deposition of DNA sequences:

Sequences are available at the NCBI SRA: PRJNA1254420, SRR33279779–SRR33279796.

## Data Availability

Raw data is available in the Supplemental Files.

## Supplemental Information

Supplemental information for this article can be found online at http://dx.doi.org/10.7717/peerj.19944#supplemental-information.

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
