# Peer review of "Effects of Undaria pinnatifida-derived brown algae polysaccharide (UPS) on the nutritional composition, digestive capacity, immune performance and intestinal microbiota of juvenile sea cucumber (Apostichopus japonicus)"

_PeerJ, doi:10.7717/peerj.19944_

## Round 0.1 · original submission · Major Revisions

There is some serious points/suggestions/amendments suggested by the reviewers that I found is very relevant. Authors need to address each of the concerns carefully before this manuscript can be accepted. This is especially on the methodology to clarify each experiments clearly, statistically tests selected and other editorial mistakes should be resolved.

Reviewer 1 ·

Basic reporting

The study addresses a relevant topic in sustainable aquaculture by investigating Undaria pinnatifida polysaccharide (UPS) as a functional feed additive for Apostichopus japonicus. The experimental design is generally robust, and the multi-parameter analysis (nutrition, digestion, immunity, microbiota) provides holistic insights. However, several key issues require revision before publication.
1- The manuscript is generally well-written, but minor language polishing would improve readability.
2- Inconsistent italicization of genus/species. Correct taxonomic ranks and use italics for genera.
3- The novelty should be improved and cite recent studies (2020–2024) on polysaccharide blends in echinoderms in the introduction.
4- Confirm that the growth rate was not affected in the abstract by adding "although growth performance remained unchanged".

Experimental design

1- whether "Microbial Brick 85" (Bacillus subtilis, Lactobacillus, yeast) could influence UPS effects? please clarify and explain why you used it.
2- Justify dosage selection with preliminary data or citations.
3- What is the weight of the animals?

Validity of the findings

1- While the manuscript attributes increased H₂O₂ to improved immune performance, it is important to recognize that H₂O₂ can also be a marker of oxidative stress when produced in excess. Please clarify.
2- Discuss why growth was unaffected despite other improvements.
3- Why did you find shifts in microbial taxa?
4- Include a comparative assessment of UPS with other known immunostimulatory polysaccharides.
5- Future research could investigate the molecular pathways through which UPS influences digestive enzymes, immune responses, and microbial populations.

·

Basic reporting

The manuscript (peerj-118383) systematically investigated the effects of UPS (Undaria pinnatifida polysaccharide) with varying concentrations on the levels of physiology and gut microbiota warranting in juvenile Apostichopus japonicus. The study is timely, of interest to sustainable aquaculture and covers a wide range of markers (nutritional, enzymatic, immune, microbiota). The whole manuscript was with logical writing order and easily to read, while some specific issues listed below should be addressed before this paper could be considered for publication.

Experimental design

Major comments:
Lines 130–134: All diets were fermented and mixed with marine sediments prior to being supplied. This also makes it challenging to differentiate UPS effects from the impact of fermentation and sediment on the microbiota. A control with unfermented feed would be necessary.
Lines 192–193: Only 3–5 replicates are reported per group. For a study with multifaceted outcomes like microbiota and immunoassays, this result would be unsatisfactory for reliable statistical evidence. The sample size should be justified by performing a power analysis or more replicates.
Lines 201–203: No significant differences in measures of growth are reported by authors, but discussion and conclusion stress growth benefits for "healthy growth". Such allegations are unsupported by data and should either be qualified or withdrawn in the absence of new evidence.
Lines 235–248: Beta diversity analysis is referred to, but little served mention of sequencing depth, rarefaction, or community evenness. Read counts and rarefaction curves need to be provided by authors and also state whether the sequencing depth was normalized.
Lines 283–297: Functional predictions using PICRUSt2 on 16S rRNA gene data should be interpreted cautiously because of their inherent limitations. Mechanistic implications of such predictions (e.g., siderophore synthesis) are speculative and risky. These should be stated as tentative hypotheses requiring additional confirmation.
Many English grammar and wording errors appear in the text, and I recommend native speaker proof-reading and correction if the authors have not already done so. For instance: Line 28: Change “are important bioactive macromolecules” into “represent bioactive macromolecules with potential functional feed applications”. Line 215: “with a 5.8-fold increase relative to HS0” is not clearly contextualized.

Validity of the findings

Minor comments:
Line 176: Assay kits were listed and enzyme reaction conditions or standards without a description were not mentioned. State reading absorbances, the incubation temperature and normalization approach.
Lines 320–334: The manuscript goes through and summarizes the results with little depth or comparison to mechanistic literature (e.g., that UPS improves nutrient content).
Several references (e.g., Lines 547–550, Lines 595–597) lack consistency in punctuation and abbreviation style. Ensure journal names are italicized and DOI format is standardized.

·

Basic reporting

This study investigated the effects of six different levels (0%, 0.5%, 1%, 1.5%, 2%, and 3%) of Undaria pinnatifida-derived polysaccharide (UPS) on the nutritional composition of the body wall, digestive capacity, immune performance, and diversity of the intestinal microbiota in juvenile sea cucumbers (Apostichopus japonicus). I read the manuscript with interest. However, in my view, the manuscript requires substantial revision. particularly in the Methodology (data analysis) and Results sections before it can be considered for publication.

Experimental design

The experimental design also lacks clarity. The experiment was conducted using six 70-L square glass tanks, each divided into three sections by partitions, with 1 kg of sea cucumber distributed per tank. However, this description is ambiguous. It is unclear whether the partitions served to create independent replicates or were simply spatial dividers. The manuscript should clearly specify the replication strategy and statistical units. Further clarification is needed regarding how the animals were distributed among the experimental units, along with information about their initial weight and length. Moreover, sample sizes for the various analyses including nutritional composition, digestive enzyme activity, immune enzyme assays, and intestinal microbiota analysis are not clearly stated. This information is essential for assessing the statistical robustness of the study.

Validity of the findings

The authors state that a one-way ANOVA was used for multi-group comparisons and that Duncan’s multiple range test was applied for post hoc analysis (lines 193–195). However, the Duncan test is known to perform poorly in controlling Type I error compared to other post hoc tests, such as the Tukey or Newman-Keuls procedures [see Day RW, Quinn GP (1989) Comparisons of treatments after an analysis of variance in ecology. Ecol. Monogr. 59: 433–463]. I recommend replacing the Duncan test with either the Tukey or Newman-Keuls test for all post hoc comparisons. This change may affect the outcomes of some statistical comparisons and could necessitate revisions in the interpretation of the results.
Moreover, there is insufficient information on other aspects of the data analysis. For instance, the methodology used to generate the flower plot of ASVs (Figure 1A) is not described. Similarly, no details are provided about the PCoA analysis (Figure 1B). The methods section should include a more comprehensive explanation of how the PCoA was conducted, including the distance metric used, ordination method, and software package or scripts.
Figure 3 presents the intestinal microbial composition, which consists of compositional (i.e., relative abundance) data. These are non-normally distributed, and thus not suitable for standard parametric tests. However, the manuscript does not specify how these data were analyzed. Additionally, information about multiple comparison correction is missing and should be included in the data analysis section.
The results section contains several inaccurate statements. For example, the claims in lines 230 and 231–232 are not supported by the data in Table 4. Similar inconsistencies are observed in other parts of the Results section. A thorough revision is necessary to ensure that all results are correctly described and accurately reflect the presented data.

---

## Round 0.2 · accepted · Accept

After careful revision and addressing all the comments from the reviewers, we are happy to accept your manuscript for publication.

Reviewer 1 ·

Basic reporting

Careful language editing is needed.

Experimental design

The authors responded to my comments, and no other comments.

Validity of the findings

The authors responded to my comments, and no other comments.

·

Basic reporting

I recommend this manuscript is acceptable for publication in present version.

Experimental design

No comments.

Validity of the findings

No comments.

Additional comments

No comments.

·

Basic reporting

I have reviewed the revised manuscript and found that most of my comments have been appropriately addressed. I recommend that the manuscript be accepted for publication.

Experimental design

OK

Validity of the findings

OK

Additional comments

None